# Pulmonary Adenocarcinoma with Cutaneous Metastasis in a Dog

**DOI:** 10.3390/vetsci11070312

**Published:** 2024-07-11

**Authors:** Anita Greyling, Louise van der Weyden, Antonia V. Lensink, Nicolize O’Dell

**Affiliations:** 1Department of Paraclinical Sciences, Faculty of Veterinary Science, University of Pretoria, Onderstepoort 0110, South Africa; 2Wellcome Sanger Institute, Wellcome Genome Campus, Hinxton, Cambridge CB10 1SA, UK; lvdw@sanger.ac.uk; 3Department of Anatomy and Physiology, Faculty of Veterinary Science, University of Pretoria, Onderstepoort 0110, South Africa; antoinette.lensink@up.ac.za

**Keywords:** canine, lung, skin, tumour, primary, metastasis, electron microscopy, microvilli

## Abstract

**Simple Summary:**

Primary lung cancer is rare in dogs and the prognosis can often be poor. In this report, we describe the post-mortem evaluation of a dog with a single large mass in one lobe of the lung, numerous smaller masses spread throughout the other lobes, and a single skin mass on the chest. Histopathological examination of the lung samples revealed multiple proliferations of neoplastic epithelial cells, with some showing microvilli-like structures on their luminal cytoplasmic edges. Histopathological examination of the skin mass revealed neoplastic cells that closely resembled those seen in the lung. Electron microscopy of the skin samples showed cells resembling the respiratory epithelium, along with cells exhibiting microvilli, which are indicative of cilia. The diagnosis was pulmonary adenocarcinoma with cutaneous metastasis, and this is the first report of a dog with lung cancer metastasizing to the skin.

**Abstract:**

Primary lung cancer is rare in dogs and depending on the tumour stage and subtype, the prognosis can be poor. In this report, we describe a 10 year-old female intact Yorkshire terrier that presented progressive weight loss and chronic pain of unknown origin. Due to the poor condition of the dog, it was subsequently euthanized. Post-mortem evaluation revealed a single large mass in the left caudal lung lobe, with numerous pale, proliferative lesions of various sizes dispersed throughout all the lobes. Additionally, a solitary skin mass was palpated on the mid-thoracic body wall. Histopathological examination of the lung samples revealed multiple distinct, non-encapsulated, expansive neoplastic epithelial cell proliferations with dense cellularity, exhibiting growth patterns, ranging from papillary to micropapillary to solid, accompanied by central areas of necrosis. In some areas, microvilli-like structures were observed on the luminal cytoplasmic margins of the neoplastic cells. The histopathology of the skin mass closely resembled that of the lung. Electron microscopy of the skin samples revealed regions containing cells resembling the respiratory epithelium, along with cells exhibiting processes or microvilli indicative of cilia. The diagnosis was pulmonary adenocarcinoma with cutaneous metastasis. This is the first report of a canine with primary lung cancer that metastasized to the skin.

## 1. Introduction

Primary lung cancer is rare in dogs, with estimates of incidence ranging from 4.2 per 10,000 dogs per year [1], to 15 per 100,000 dogs per year [2], to 0.1–0.9% of dogs [3]. Of primary lung tumours, the most common subtype in dogs is pulmonary carcinoma (PPC). Among PPC cases, the most common histological subtype is pulmonary adenocarcinoma, reported to account for 60–80% of cases [3,4].

PPC is a disease mostly of older dogs [3,5,6]. There is no apparent sex predisposition; however, some studies have reported higher incidences of PPC in particular breeds, including the Boxer, Doberman Pinscher, Irish Setter, Australian Shepherd, and Bernese Mountain Dog, although other studies do not find any breed bias [3,5,6,7].

PPCs can be very aggressive tumours, with one study finding that ~50% of cases showed invasive features or metastatic lesions in adjacent lung lobes or bronchial lymph nodes at the time of diagnosis [8]. Another study of malignant lung tumours in dogs found that 34% had vascular, lymphatic, or intrapulmonary spread, 13% had local lymph node metastases, and 23% had distant metastases [6]. Lung cancer metastasis in dogs most commonly affects other lobes of the lung or lymph nodes, followed by the bone and brain. However, other sites include thoracic cavity organs and tissues, liver, pancreas, adrenal gland, and kidney. There have been individual case reports of the metastasis of primary pulmonary adenocarcinoma to the uvea, brain and adrenal gland [9], and brain stem [10]. This is the first report of a cutaneous metastasis of a pulmonary carcinoma in dogs.

## 2. Case Report

A 10 year-old female intact Yorkshire terrier was presented to the Onderstepoort Veterinary Academic Hospital with chronic pain of unknown origin and progressive weight loss, weighing only 0.86 kg. Due to the financial constraints of the owner, as well as the chronic nature of the disease together with a suspect intra-abdominal mass on the palpation, the decision was made to humanely euthanize the dog. External post-mortem evaluation revealed moderate dehydration and pale mucous membranes. The body condition score was poor, with muscle atrophy, body prominences, and decreased fat reserves (Figure 1a). Incidentally, the jaw could not fully close due to an ingested lamb bone lodged over the left carnassial tooth.

A single, well-marginated skin mass was present in the mid-thorax of the right body wall (Figure 1b). A blood smear revealed a mild monocytosis, mild neutrophilia, and mild to moderate anisocytosis. Faecal analysis revealed Strongyloid-type ova and nematodes in the small intestine.

Upon opening the thoracic cavity, multiple widespread, pale proliferative lesions were observed in the lung parenchyma. The left caudal lung lobe had one large mass (8 cm × 8 cm) and smaller nodules of varying sizes were observed in the rest of the lobes (Figure 1c). Sectioning of the larger mass revealed a necrotic centre containing a thick, yellowish-green fluid (Figure 1d). Incidental findings included mild hydrometra and anthracosis.

Tissue impression smears of the lung and cutaneous neoplastic masses were made and stained using the Diff-Quick staining protocol. Cytological evaluation for both samples revealed rafts of neoplastic epithelial cells (Figure 2a) displaying a moderate degree of pleomorphism, single prominent nucleoli, and occasional mitoses (Figure 2b). Scattered neutrophils and necrotic cellular debris were occasionally observed.

Tissue samples were taken from the pulmonary masses and the cutaneous mass, fixed in 10% buffered formalin, and routinely processed for histopathological evaluation. Sections (4 μm) were stained with haematoxylin and eosin (HE) and examined by light microscopy by a veterinary pathologist. Histological analysis of the lung lesions revealed multiple well-demarcated, non-encapsulated, expansile, dense cellular proliferations of neoplastic epithelial cells forming various growth patterns. The growth patterns ranged from papillary (Figure 3a) to micropapillary (Figure 3b), to solid with areas of central necrosis (Figure 3c). The neoplastic cells displayed a moderate degree of pleomorphism, ranging from polygonal to columnar and cuboidal with indistinct cytoplasmic margins. In some areas, the luminal cytoplasmic margins of the neoplastic cells displayed what appeared to be microvilli. The nuclei were round to oval, with vesicular chromatin mostly containing a single magenta nucleolus, and the mitotic rate averaged five mitoses per high-power field. Lymphovascular invasion by the neoplastic cells was evident. The histopathology of the skin mass appeared almost identical to that observed in the lung, with some of the cells appearing to have microvilli on their surface (Figure 3d) and a mitotic rate of fifty mitoses in 2.37 mm^2^ (10 contiguous fields of view under 40× magnification). In addition, lymphovascular invasion was also evident.

Tissue samples were taken from the formalin-fixed skin lesion, post-fixed in 1% osmium tetroxide in Millonig’s buffer, and routinely processed for standard transmission electron microscopy (TEM) to confirm the presence of microvilli and their pulmonary origin. Ultra-thin resin sections were stained with uranyl acetate and lead citrate, and examined by TEM; however, the preservation of the sample was insufficient for observing the interior microtubular structure of the microvilli, which is required to demonstrate the presence of ciliated epithelium. Nonetheless, regions containing cells with morphological traits comparable to respiratory epithelia, as well as cells with microvilli that were presumed to be cilia, were found (Figure 4). Based on the histological similarities between the lung and skin masses and the electron microscopy results, a diagnosis of primary pulmonary adenocarcinoma with cutaneous metastasis was favoured.

## 3. Discussion

Primary lung cancer is the most common diagnosed cancer in humans, and the leading cause of cancer-related deaths [11]. In contrast, primary lung cancer is rare in canines [1,2,3]. However, in both species, primary lung cancer is typically a disease of older individuals, with the average age of diagnosis being 70 years in humans [12] and 10–11 years in canines [3,5,6], which is consistent with the age of the dog in this study (10 years). Similar to humans, PPC in dogs can present as a single growth or multifocal growths. Where single growths are present, they have been observed more frequently in the right lobes, particularly in the caudal lung [4]. Multifocal growths are fairly common, with several necropsy surveys reporting ~40% of cases showing PPC in one or more lobes [4], as is the case for the dog in this study. These multifocal growths may represent intrapulmonary metastases, which can occur via lymphatic routes or via bronchial invasion and re-aspiration with intra-airway seeding into the other lobes [4].

A staging system for canine lung tumours was proposed by the World Health Organization in 1980, and in 2020, the canine lung carcinoma stage classification (CLCSC) was proposed, adapted from the human lung cancer stage classification [13]. Primary tumour features (T1–T4), tumour grade (Stage 1–4), tumour size, lymph node metastasis status (N0–N2), and distant metastasis status (M0–M1) have been shown to be significant prognostic factors for both median survival time (MST) and overall survival (OS) [13,14]. The MST of dogs with primary lung tumours showing no clinical signs is 545 days; however, the MST for those with clinical signs (most frequent being coughing, followed by dyspnea, lethargy, weight loss, and tachypnea [3]) is only 240 days [15]. The dog in this report presented chronic pain and weight loss, and due to the severity of her condition, she was euthanised immediately.

The distinction of primary lesions from metastatic lesions is a critical step in the diagnosis of neoplasia in the lung, as metastatic lesions are more common than PPC in dogs [4]. In this case report, the presence of a skin lesion did raise the possibility of the lung masses being metastatic lesions. Indeed, differential diagnoses of apocrine adenoma or apocrine adenocarcinoma were considered due to the location of the skin mass and the glandular histopathological features observed. However, the macroscopic characteristics of the tumour distribution, specifically a large mass surrounded by smaller nodules, supported the diagnosis of primary lung neoplasia [4]. In addition, TEM of the skin mass revealed regions of neoplastic cells with morphological traits comparable to respiratory epithelium, as well as cells with processes or microvilli that were presumed to be cilia. Indeed, electron microscopy has been previously used to confirm the adenocarcinoma origin of a lung tumour in a dog, with the neoplastic cells showing characteristic features of glandular cells with microvilli, numerous free ribosomes, large round secretory granules, and intercellular desmosomes [16].

Currently, the primary treatment for PPCs is complete surgical excision when possible. However, in the advanced (metastatic) disease setting, prognosis is poor even if surgery is pursued, with 40–50% experiencing disease progression [13,14,15]. Studies have reported trialling adjuvant (post-surgery) chemotherapy, including vinorelbine, carboplatin (single agent), alternating vinorelbine and carboplatin, and alternating vinblastine and cisplatin [13,14,17] in dogs with PPC. However, in general, the OS is poor for dogs with advanced-stage disease, and to-date the use of adjuvant chemotherapy in dogs with PPC has not demonstrated any benefit in MST or OS [13,14,17,18]. For dogs with unresectable or advanced PPC, where surgery is contraindicated, metronomic chemotherapy (comprising low-dose cyclophosphamide, piroxicam, and thalidomide) has shown some benefits, with a MST of 129 days versus 60 days in dogs with no treatment [19]. More recently, a study assessing the benefits of vinorelbine as a first-line treatment in dogs with Stage 4 PPC found it was well-tolerated with an adequate toxicity profile, and a partial response was documented in 8/10 dogs [20]. Additional studies are needed to follow up on the potential management of advanced PPC in canines using vinorelbine as first-line treatment. Recently, novel canine pulmonary adenocarcinoma (PAC) cell lines have been generated, which may prove useful in future research for the development of treatments for canine PAC [21]. Indeed, an approach being considered is oncolytic virotherapy, with a recent study demonstrating anti-tumour activity of a recombinant measles virus against canine lung cancer cell lines in vitro [22]. Of course, future studies, including clinical trials of this treatment, will be needed to assess its in vivo efficacy.

The aetiology of lung cancer is dogs in not known. As second-hand tobacco smoke [23] and inhalation of indoor or outdoor polluted air is associated with primary lung cancer in people [24,25], a logical assumption would be that passive cigarette smoking and/or urban living may play a role in increasing the risk of lung tumours in dogs. Indeed, one study detected cotinine in the urine of dogs exposed to cigarette smoke, as well as an increase in the macrophage and lymphocyte populations, and the presence of anthracosis (black dust matter) in the cytoplasm of alveolar macrophages, confirming environmental tobacco exposure [26]. A higher incidence of PPC has been reported in small-sized dogs (postulated to possibly be as a result of spending more time indoors, in close contact with their owners, thereby possibly being exposed to carcinogens) [26] and brachycephalic breeds (postulated to possibly be due to reduced nose filtration ability) [27]. However, a definitive association between second-hand tobacco smoke and lung cancer in dogs has not yet been found [27,28]. Nevertheless, it is possible that tobacco smoke exposure plays a role in the development of PPC in a sub-group of dogs, such as tumours of certain histologic types and/or dogs of a specific breed/nose length, and further studies with a much larger number of dogs is necessary. Interestingly, anthracosis was observed in the dog in this report, and a study has reported an association between anthracosis, due to inhalation of polluted air, and lung cancer in dogs [29]. Furthermore, the amount of anthracosis has been shown to correlate with the relative proportion of the lung tumour cells expressing EGFR, with a trend towards shortened survival for the high EGFR group [29]. Another study found that exposure to household radon might play a role in the development of primary pulmonary neoplasia in dogs, with approximately a two-fold increase in incidence of primary pulmonary neoplasia in dogs living in ‘high radon zone’ areas compared to ‘low radon zone’ areas [30].

In humans, cutaneous metastases occur in 1–12% of lung cancers, with adenocarcinomas being the most common histological subtype [31]. In 20–60% of cases the skin lesions present before or synchronously with the diagnosis of the primary lung tumour [31] and the MST for patients with cutaneous metastasis of lung cancer is 5–6 months [31]. In canines, lung cancer metastasis most commonly affects additional lung lobes or lymph nodes, although metastasis to other organs has been reported including the bone, brain, thoracic cavity organs and tissues, liver, pancreas, adrenal gland, and kidney. However, this is the first report of cutaneous metastasis of a lung cancer in dogs.

## 4. Conclusions

This report serves to show further parallels between cancer in canines and humans, and thus is important from a comparative oncology point of view. It also serves to increase our understanding of lung cancer in dogs by demonstrating that the presence of a cutaneous mass may be the first indication of this disease, and thus is diagnostically important. It is only through a thorough understanding of lung cancer in dogs that we can hope to address the unmet clinical need in preventing and effectively managing canine PPC.

## Figures and Tables

**Figure 1 vetsci-11-00312-f001:**
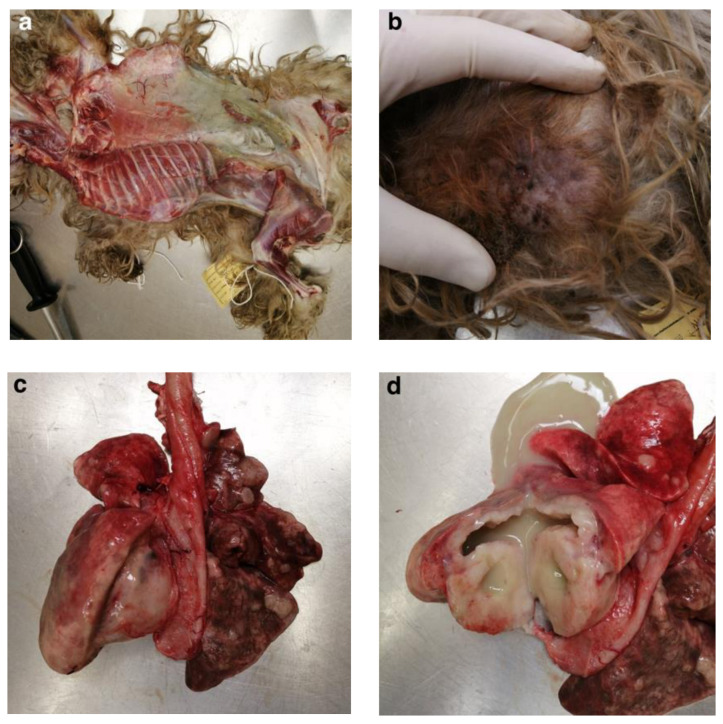
Macroscopic findings at post-mortem. (**a**) Poor body condition with muscle atrophy and bony prominences. (**b**) Right thoracic cutaneous mass. (**c**) Lung with one large neoplastic mass in the left caudal lobe and multiple widespread smaller neoplastic masses. (**d**) Necrotic centre of the primary neoplastic mass in the left caudal lung lobe.

**Figure 2 vetsci-11-00312-f002:**
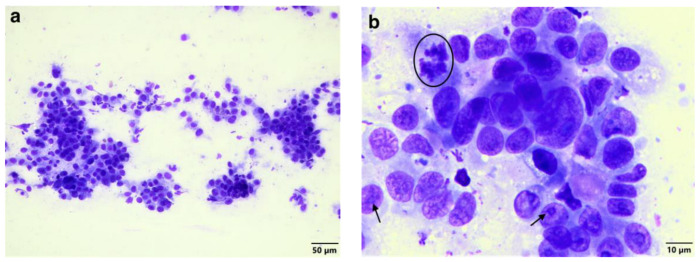
Impression smear of the cutaneous mass. (**a**) Rafts of neoplastic epithelial cells (Diff-Quick stain, 200× magnification). (**b**) Neoplastic epithelial cells showing a moderate degree of pleomorphism, single prominent nucleoli (*arrows*), and occasional mitoses (*circle*) (Diff-Quick stain, 1000× magnification).

**Figure 3 vetsci-11-00312-f003:**
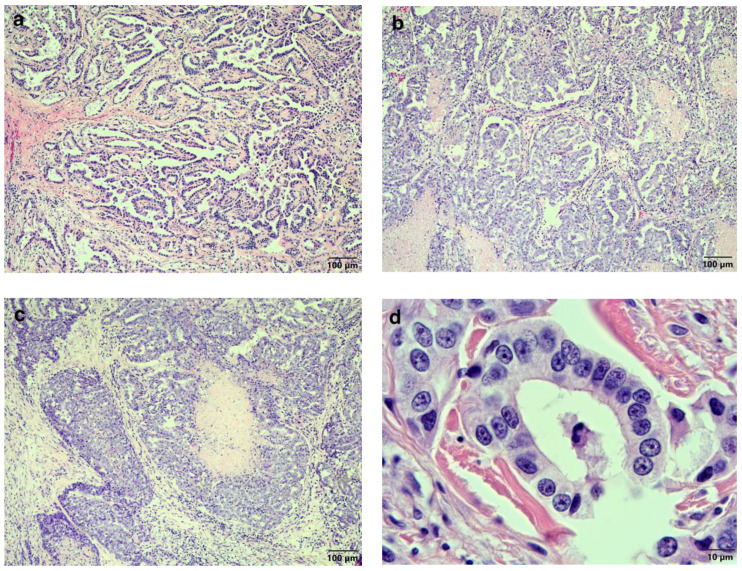
Histopathological findings of the pulmonary and cutaneous lesions. The pulmonary lesions were composed of multiple, well-demarcated, non-encapsulated, expansile, densely-cellular proliferations of neoplastic epithelial cells, that showed growth patterns ranging from (**a**) papillary (HE stain, 100× magnification) to (**b**) micropapillary (HE stain, 200× magnification), to (**c**) solid with areas of central necrosis (HE stain, 100× magnification). (**d**) The neoplastic cells of the cutaneous mass appeared to have microvilli on the luminal cytoplasmic margins (HE stain, 1000× magnification).

**Figure 4 vetsci-11-00312-f004:**
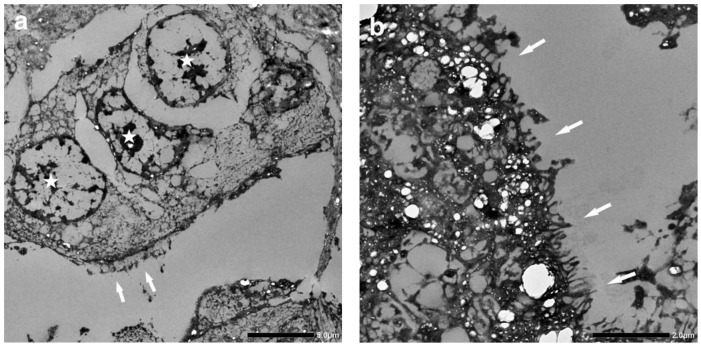
Ultrastructural findings of the skin lesion. Transmission electron microscopy of the skin lesion showed (**a**) areas of suspected ciliated (*arrows*) columnar epithelial cells (*stars*) (scale bar = 5 µm). (**b**) Higher magnification of the suspected cilia/microvilli (*arrows*) (scale bar = 2 µm).

## Data Availability

All data underlying the results are available as part of the article and no additional source data are required.

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
