# Peer review of "Pulmonary Adenocarcinoma with Cutaneous Metastasis in a Dog"

_vetsci, 2024, doi:10.3390/vetsci11070312_

Round 1

Reviewer 1 Report

Comments and Suggestions for Authors

Thank you for submitting your manuscript for review. This case report is prepared correctly in accordance with the rules for this type of reports. The description of the morphology is supported by very good quality microphotographs.

After reading the manuscript, I have some minor comments:

- Simple Summary and Abstract are significantly similar in terms of content

- some macroscopic images relating to the general health and maintenance of the dog seem unnecessary to me, e.g. Fig. 1a, b, c, d.

- understanding the financial conditions of the owner, however, it seems to me that this is not enough to make a decision of ethanasia. Chronic pain is treatable. The diagnostic procedure did not include chest X-ray and abdominal ultrasound. In other words, the dog was euthanized at the owner's request. Is it in line with welfare principles?

- line 120. According to ECVP/ESVP recommendations, the number of mitoses should be given in 10 fields of view under 40x magnification with a specific area e.g. 2.37mm2 (Meuten DJ, Moore FM, George JW. Mitotic Count and the Field of View Area: Time to Standardize. Vet Pathol. 2016 Jan;53(1):7-9. doi: 10.1177/0300985815593349.

Reviewer 2 Report

Comments and Suggestions for Authors

In the publication, it was mentioned that the dog had the bony protrusion in its jaw. Was it known if this was congenital or was it related to the cancer at all?

On line 238, it is mentioned that human lung cancers express EGFR. Any chance the tissue sections obtained from this dog could be stained to see if it expresses EGFR similar to humans?

Reviewer 3 Report

Comments and Suggestions for Authors

Dear authors,

 Generally case reports describe clinical cases observed in clinical care settings and are particularly useful to ascertain the effectiveness of treatment in the individual patient, detect adverse effects, identify new diseases, recognise clinical presentations of rare diseases or unusual presentations of common diseases.

 In your case you highlight a clinical presentation that is rather rare, and as such often misdiagnosed.

The case is well described and argued, the figures are of good quality, very nice the ones at TEM.

The discussion is well structured.

As far as I am concerned, and for my expertise, I have no suggestions for change.

Greetings
